# Dirichlet belief networks for topic structure learning

**He Zhao[1], Lan Du[1]\* Wray Buntine[1], and Mingyuan Zhou[2]\***
[1]Faculty of Information Technology, Monash University, Australia
[2]McCombs School of Business, The University of Texas at Austin, USA

## Abstract

Recently, considerable research effort has been devoted to developing deep architectures for topic models to learn topic structures. Although several deep models have been proposed to learn better topic proportions of documents, how to leverage the benefits of deep structures for learning word distributions of topics has not yet been rigorously studied. Here we propose a new multi-layer generative process on word distributions of topics, where each layer consists of a set of topics and each topic is drawn from a mixture of the topics of the layer above. As the topics in all layers can be directly interpreted by words, the proposed model is able to discover interpretable topic hierarchies. As a self-contained module, our model can be flexibly adapted to different kinds of topic models to improve their modelling accuracy and interpretability. Extensive experiments on text corpora demonstrate the advantages of the proposed model.

## 1 Introduction

Understanding text has been an important task in machine learning, natural language processing, and data mining. Text is discrete, unstructured, and often highly sparse. A popular way of analysing texts is to represent them as a set of latent factors via topic modelling or matrix factorisation. With great success in modelling text, probabilistic topic models discover a set of latent topics from a collection of documents. Those topics, as latent factors, can be interpreted by distributions over words and used to derive low dimensional representations of the documents. Specifically, most existing topic models are built on top of the following generative process: Each topic is a distribution over the words (i.e., *word distribution*, WD) in the vocabulary; each document is associated with a *topic proportion* (TP) vector; and a word in a document is generated by first drawing a topic according to the document's TP, then sampling the word according to the topic's WD.

In a Bayesian setting, TPs and WDs are both imposed on prior distributions. For example, one commonly-used prior for TP and WD is a Dirichlet distribution, as in Latent Dirichlet Allocation (LDA) (Blei et al., 2003). Recently, deep hierarchical priors, especially imposed on TPs, have been developed to generate hierarchical document representations as well as discover interpretable topic hierarchies. For example, there are hierarchical tree-structured constructions based on the Dirichlet Process (DP) or Chinese Restaurant Process (CRP), such as the nested CRP (nCRP) (Blei et al., 2010) and the nested hierarchical DP (Paisley et al., 2015); deep constructions based on restricted Boltzmann machines and neural networks such as the Replicated Softmax Model (RSM) (Hinton and Salakhutdinov, 2009), the Neural Autoregressive Density Estimator (NADE) (Larochelle and Lauly, 2012), and the Over-replicated Softmax Model (OSM) (Srivastava et al., 2013); models based on variational autoencoders (VAE) including Srivastava and Sutton (2017); Miao et al. (2017); Zhang et al. (2018). Recently, models that generalise the sigmoid belief network (Hinton et al., 2006) have been proposed, such as Deep Poisson Factor Analysis (DPFA) (Gan et al., 2015), Deep Exponential Families (DEF) (Ranganath et al., 2015), Deep Poisson Factor Modelling (DPFM) (Henao et al., 2015), and Gamma Belief Networks (GBNs) (Zhou et al., 2016).

---

Compared with the considerable interest in deep models on TPs, to our knowledge, the counterparts on WDs have not been fully investigated. In this paper, we propose a new multi-layer generative process on WDs, as a self-contained module and an alternative to the single-layer Dirichlet prior. In the proposed model, WDs are the output units of the bottom layer in a DBN with hidden layers parameterised by Dirichlet-distributed hidden units and connected with gamma-distributed weights. Specifically, each Dirichlet unit in a hidden layer is a probability distribution over the words in the vocabulary and can be view as a "hidden" topic. In each layer, the Dirichlet prior of a topic is a mixture of the topics in the layer above. As the hidden units are drawn from Dirichlet, the proposed model is named the Dirichlet Belief Network, hereafter referred to as *DirBN*[2].

Compared with existing related deep models, DirBN has the following appealing properties: **1) Interpretability of hidden units**: Every hidden unit in every layer of DirBN is a probability distribution over the words, making them real topics that can be directly interpreted. **2) Discovering topic hierarchies**: The mixture structure of DirBN enables the model to enjoy a straightforward way of discovering semantic correlations of topics in two adjacent layers, which further form topic hierarchies with the multi-layer construction of the model. Due to the intrinsic abstraction effect of DBN, the topics in the higher layers are more abstract and can be treated as the generalisation of the ones in the lower layers. **3) Better modelling accuracy**: It is known that TPs are local variables (specific to individual document), while WDs are global variables over the target corpus. Unlike many other hierarchical parallels on TP, DirBN imposes a deep structure on WD, which "absorbs the information" from the entire corpus. It makes DirBN be able to get better modelling accuracy especially in the case of sparse texts such as tweets and news abstracts, where the context information of an individual document is not enough to learn a good model using existing approaches. **4) Adaptability**: As many sophisticated models on TPs usually use a simple Dirichlet prior on WDs, including the well-known ones such as Supervised Topic Model (Mcauliffe and Blei, 2008) and Author Topic Model (Rosen-Zvi et al., 2004), our DirBN can be easily adapted to them to further improve modelling accuracy and interpretability.

In conclusion, the contributions of this paper include: **1)** We propose DirBN, a deep structure that can be used as an advanced alternative to the Dirichlet prior on WDs with better modelling performance and interpretability. **2)** We demonstrate our model's adaptability by applying DirBN with several well-developed models, including Poisson Factor Analysis (PFA) (Zhou et al., 2012), MetaLDA (Zhao et al., 2017a), and GBN (Zhou et al., 2016). **3)** With proper data augmentation and marginalisation techniques, DirBN enjoys full local conjugacy, which facilitates the derivation of a simple and effective inference algorithm.

## 2 The proposed DirBN

In this section, we introduce the details of the generative and inference processes of DirBN.

### 2.1 Generative process

We first define the essential notation and review the basic framework of topic modelling, followed by the details of the proposed DirBN. Assume that the bag-of-words of document $d$ in a corpus with $N$ documents and $V$ unique words in the vocabulary are stored in a count vector $\boldsymbol{x}_d \in \mathbb{N}_0^V$. A topic model with $K$ topics is composed of the TP vector $\boldsymbol{\theta}_d \in \mathbb{R}_+^K$ for each document $d$ and the WD vector $\boldsymbol{\phi}_k \in \mathbb{R}_+^V$ for each topic $k$ ($k \in \{1, \cdots, K\}$). To generate a word in document $d$, one can first sample a topic according to its TP, and then sample the word type according to the topic's WD. Given this framework, many prior constructions of TPs have been proposed, such as the Dirichlet distribution in LDA, logistic normal distributions for modelling topic correlations in Correlated Topic Model (CTM) (Lafferty and Blei, 2006), nonparametric priors like the Hierarchical Dirichlet Process (Teh et al., 2012), and recently-proposed deep models like DPFA (Gan et al., 2015), DPFM (Henao et al., 2015), and GBN (Zhou et al., 2016). Unlike the extensive choices for constructing TP, the symmetric Dirichlet distribution on WDs still dominates in many advanced topic models. Here DirBN is a new hierarchical approach of constructing WDs, detailed as follows.

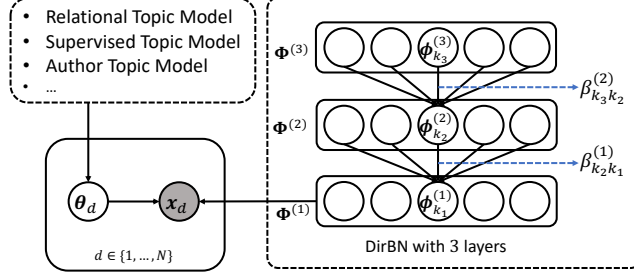

Figure 1: Demonstration of the generative process of DirBN with three layers.

A DirBN with $T$ layers leaves the TPs of the basic framework untouched and draws $\phi_k$ according to the following generative process:

$$\phi_{k_T}^{(T)} \sim \text{Dir}_V(\eta),$$
$$\cdots$$
$$\phi_{k_t}^{(t)} \sim \text{Dir}_V(\psi_{k_t}^{(t)}), \psi_{k_t}^{(t)} = \sum_{k_{t+1}}^{K_{t+1}} \phi_{k_{t+1}}^{(t+1)} \beta_{k_{t+1}k_t}^{(t)}, \beta_{k_{t+1}k_t}^{(t)} \sim \text{Ga}(\gamma_{k_{t+1}}^{(t)}, 1/c^{(t)}),$$
$$\cdots$$
$$\phi_{k_1}^{(1)} \sim \text{Dir}_V(\psi_{k_1}^{(1)}), \ \psi_{k_1}^{(1)} = \sum_{k_2}^{K_2} \phi_{k_2}^{(2)} \beta_{k_2 k_1}^{(1)}, \beta_{k_2 k_1}^{(1)} \sim \text{Ga}(\gamma_{k_2}^{(1)}, 1/c^{(1)}), \quad (1)$$

where 1) $\text{Ga}(-,-)$ is the gamma distribution with shape and scale parameters and $\text{Dir}_V(-)$ is the Dirichlet distribution[3]; 2) The superscript with a bracket over a variable indicates which layer it belongs to and $k_t \in \{1, \cdots, K_t\}$ is the topic index in the $t$-th layer; 3) The output of DirBN is $\phi_{k_1}^{(1)}$, which corresponds to $\phi_k$ in the basic framework and hereafter, we use $\phi_{k_1}^{(1)}$ instead; 4) We further impose gamma priors on the following variables: $\eta \sim \text{Ga}(a_0, 1/b_0)$, $\gamma_{k_{t+1}}^{(t)} \sim \text{Ga}(\gamma_0^{(t)}/K_t, 1/c_0^{(t)})$, $\gamma_0^{(t)} \sim \text{Ga}(e_0, f_0)$, $c_0^{(t)} \sim \text{Ga}(g_0, 1/h_0)$, and $c^{(t)} \sim \text{Ga}(g_0, 1/h_0)$. The generative process of a topic model equipped with DirBN is demonstrated in Figure 1.

The idea of our DirBN can be summarised as follows:

1. From a bottom-up view, DirBN is a multi-layer matrix factorisation, which factorises the matrix of the WDs in the $t$-th layer as: $\Phi^{(t)} \sim \text{Dir}(\Phi^{(t+1)}\mathbf{B}^{(t)})$. Here we define $\Phi^{(t)} \in \mathbb{R}_+^{V \times K_t}$ ($\phi_{k_t}^{(t)}$ is the $k_t$-th column) and $\mathbf{B}^{(t)} \in \mathbb{R}_+^{K_{t+1} \times K_t}$ ($\beta_{k_t}^{(t)}$ is the $k_t$-th column). From a top-down view, the model can be considered as a stochastic feedforward network (Tang and Salakhutdinov, 2013), where the input matrix in $\Phi^{(T)}$, the output matrix is $\Phi^{(1)}$, and the stochastic units are drawn from the Dirichlet distribution.

2. As DirBN is a Bayesian probabilistic model, consider a DirBN with only two layers as an example: each first-layer topic $\phi_{k_1}^{(1)}$ is drawn from a Dirichlet with the topic-specific asymmetric parameter $\psi_{k_1}^{(1)}$, which is a mixture of the second-layer topics. So the statistical strength is shared via the mixture, which plays an important role in handling sparse texts.

3. In DirBN, not only in the bottom layer, but also in any other layer $t$, each hidden unit is a distribution over the vocabulary and can be viewed as *real topic* directly interpreted by words. Although the bottom layer serves as the actual WDs for generating the words, the topics in the higher layers are involved with the belief prorogation in the network.

4. The weight $\beta_{k_{t+1}k_t}^{(t)}$ is drawn from a hierarchical gamma prior (i.e., the shape parameter $\gamma_{k_{t+1}}^{(t)}$ of the gamma prior on $\beta_{k_{t+1}k_t}^{(t)}$ is also drawn from a gamma). It allows topics in the $(t+1)$-th layer to contribute differently to those in the $t$-th layer. In addition, the hierarchical structure on $\beta_{k_{t+1}k_t}^{(t)}$ is similar to the one in Zhou (2015), which provides an

intrinsic shrinkage mechanism on $\boldsymbol{\beta}_{k_t}^{(t)}$. In other words, each $k_t$ is expected to be sparsely connected by a subset of $k_{t+1}$. We will demonstrate the shrinkage effect of DirBN in the experiments.

## 2.2 Inference process

The learning of DirBN can be done by the inference of its latent variables, i.e., $\boldsymbol{\Phi}^{(t)}$ and $\mathbf{B}^{(t)}$ for all $t$. With several data augmentation techniques, we are able to derive a layer-wise Gibbs sampling algorithm facilitated by local conjugacy. Given $\theta$ and $\phi$ (despite their constructions), a topic model usually samples the topic assignment of each word in the corpus. After that, each topic $k_1$ is associated with a vector of word counts, denoted as $\boldsymbol{x}_{k_1}^{(1)} = [x_{1k_1}^{(1)}, \cdots, x_{Vk_1}^{(1)}]$, which encodes the semantic information of topic $k_1$ and is one of the *input count vectors* of DirBN in the inference process. Given the input vectors, the inference of DirBN involves two key steps: **1)** propagating the semantic information of the input vectors up to the top layer *via* latent counts; **2)** updating $\boldsymbol{\Phi}^{(t)}$ and $\mathbf{B}^{(t)}$ down to the bottom given the latent counts. Without loss of generality, we illustrate the inference details with a two-layer DirBN as follows[4]:

**Propagating the latent counts from the bottom up**  By integrating $\phi_{k_1}^{(1)}$ out from its multinomial likelihood, we can get the likelihood of $\boldsymbol{\psi}_{k_1}^{(1)}$ as:

$$\mathcal{L}\left(\boldsymbol{\psi}_{k_1}^{(1)}\right) \propto \frac{\Gamma(\psi_{\cdot k_1}^{(1)})}{\Gamma(\psi_{\cdot k_1}^{(1)} + x_{\cdot k_1}^{(1)})} \prod_v^V \frac{\Gamma(\psi_{vk_1}^{(1)} + x_{vk_1}^{(1)})}{\Gamma(\psi_{vk_1}^{(1)})}, \tag{2}$$

where $\Gamma(-)$ is the gamma function, $\psi_{\cdot k_1}^{(1)} = \sum_v^V \psi_{vk_1}^{(1)}$, and $x_{\cdot k_1}^{(1)} = \sum_v^V x_{vk_1}^{(1)}$. By integrating $\phi_{k_1}^{(1)}$ out and introducing two auxiliary variables $q_{k_1}^{(1)}$ and $y_{vk_1}^{(1)}$, Eq. (2) can be augmented as (Zhao et al., 2017a):

$$\mathcal{L}\left(\boldsymbol{\psi}_{k_1}^{(1)}, q_{k_1}^{(1)}, y_{vk_1}^{(1)}\right) \propto \prod_v^V \left(q_{k_1}^{(1)}\right)^{\psi_{vk_1}^{(1)}} \left(\psi_{vk_1}^{(1)}\right)^{y_{vk_1}^{(1)}}, \tag{3}$$

where $q_{k_1}^{(1)} \sim \text{Beta}(\psi_{\cdot k_1}^{(1)}, x_{\cdot k_1}^{(1)})$ and $y_{vk_1}^{(1)} \sim \text{CRT}\left(x_{vk_1}^{(1)}, \psi_{vk_1}^{(1)}\right)$. Here CRT stands for the Chinese Restaurant Table distribution (Zhou and Carin, 2015; Zhao et al., 2017b). Now we can define $\boldsymbol{y}_{k_1}^{(1)} = [y_{1k_1}^{(1)}, \cdots, y_{Vk_1}^{(1)}]$, the *latent count vector* derived from the input count vector $\boldsymbol{x}_{k_1}^{(1)}$.

With $\psi_{vk_1}^{(1)} = \sum_{k_2}^{K_2} \phi_{vk_2}^{(2)} \beta_{k_2 k_1}^{(1)}$, we can then distribute the latent count $y_{vk_1}^{(1)}$ on $\psi_{vk_1}^{(1)}$ to each second layer topic $k_2$ by:

$$\left(z_{v1k_1}^{(1)}, \cdots, z_{vK_2 k_1}^{(1)}\right) \sim \text{Mult}\left(y_{vk_1}^{(1)}, \frac{\phi_{v1}^{(2)} \beta_{1k_1}^{(1)}}{\psi_{vk_1}^{(1)}}, \cdots, \frac{\phi_{vK_2}^{(2)} \beta_{K_2 k_1}^{(1)}}{\psi_{vk_1}^{(1)}}\right), \tag{4}$$

where $z_{vk_2 k_1}^{(1)}$ is the latent count allocated to $k_2$ and $\sum_{k_2}^{K_2} z_{vk_2 k_1}^{(1)} = y_{vk_1}^{(1)}$.

We now note $\boldsymbol{x}_{k_2}^{(2)} = [x_{1k_2}^{(2)}, \cdots, x_{Vk_2}^{(2)}]$ where $x_{vk_2}^{(2)} = \sum_{k_1}^{K_1} z_{vk_2 k_1}^{(1)}$. $\boldsymbol{x}_{k_2}^{(2)}$ can be viewed as one of the *output count vectors* of the first layer and also the input count vector of the second layer topic $k_2$.

In conclusion, to propagate the semantic information from the first to the second layer, we fist derive $y_{vk_1}^{(1)}$ from $x_{vk_1}^{(1)}$, then distribute $y_{vk_1}^{(1)}$ to all the second layer topics (i.e., $z_{vk_2 k_1}^{(1)}$), and finally aggregate $z_{vk_2 k_1}^{(1)}$ into $x_{vk_2}^{(2)}$.

**Updating the latent variables from the top down**  After the latent counts are propagated, we start updating the latent variables from the top layer (i.e. the second layer here). Given $\boldsymbol{x}_{k_2}^{(2)}$, $\phi_{k_2}^{(2)}$ is easy to sample from its Dirichlet posterior. With $z_{vk_2 k_1}^{(1)}$ and $\sum_v^V \phi_{k_2,v}^{(2)} = 1$, we can sample $\beta_{k_2 k_1}^{(1)}$ from its gamma posterior given the following likelihood:

$$\mathcal{L}\left(\beta_{k_2 k_1}^{(1)}\right) \propto e^{-\beta_{k_2 k_1}^{(1)}(-\log q_{k_1}^{(1)})} (\beta_{k_2 k_1}^{(1)})^{z_{\cdot k_2 k_1}^{(1)}}, \tag{5}$$

where $z^{(1)}_{\cdot k_2 k_1} = \sum_v^V z^{(1)}_{v k_2 k_1}$. Given the newly sampled $\phi^{(2)}_{k_2}$ and $\beta^{(1)}_{k_2 k_1}$, we can recompute $\psi^{(1)}_{k_1}$ and sample $\phi^{(1)}_{k_1}$ from its Dirichlet posterior. Now the inference of a two-layer DirBN is done.

## 3   Using DirBN in topic modelling

DirBN is a self-contained module on $\phi$, leaving $\theta$ untouched. Therefore, it can be used as an alternative to the simple Dirichlet prior on $\phi$ in many existing models. The adaptability of DirBN enables us to easily apply it to advanced models so that those models can benefit from the advantages of DirBN. To demonstrate this, we adapt the proposed DirBN structure to the following models:

**PFA+DirBN**   Poisson Factor Analysis (PFA) is a popular framework for topic analysis (DPFA (Gan et al., 2015), DPFM (Henao et al., 2015), GBN (Zhou et al., 2016) can be viewed as a deep extension to PFA). Specifically, we use the Bayesian nonparametric version of PFA named BGGPFA (Zhou et al., 2012), where $\theta_d$ is constructed from a negative binomial process and $\phi_k$ is drawn from a Dirichlet distribution. Note that there are close relationships between PFA and LDA, and between BGGPFA and HDP (Teh et al., 2012), analysed in Zhou (2018). Here we replace the Dirichlet construction on $\phi$ with DirBN, yielding a model named PFA+DirBN.

**MetaLDA+DirBN**   MetaLDA (Zhao et al., 2017a, 2018a) is a supervised topic model that is able to incorporate document labels to inform the learning of $\theta_d$. Keeping the structure on $\theta$ untouched, we replace the MetaLDA's structure on $\phi$ with our DirBN to get a combined model that discovers the topic hierarchies informed by the document labels. The proposed model is able to discover the correlations between labels and topic hierarchies.

**GBN+DirBN**   Recall that GBN (Zhou et al., 2015, 2016) imposes a hierarchical structure on $\theta$, which is able to learn multi-layer document representations and topic hierarchies. Here we combine DirBN and GBN together to yield a "dual" deep model, where the GBN part is on $\theta$ and the DirBN part is on $\phi$. Both parts discover topic hierarchies and the bottom-layer topics are shared by the two parts/hierarchies. It would be interesting to see how the two deep structures interact with each other.

## 4   Related work

As the proposed model introduces a hierarchical architecture on WDs (i.e., $\phi$) in topic models, we first review various priors on $\phi$, starting with the ones on sampling/optimising the Dirichlet parameters in topic models. The Dirichlet parameters in topic models were studied comprehensively in Wallach et al. (2009), which showed that Dirichlet with a symmetric parameter sampled from an uninformative gamma is the best choice. Actually, our DirBN can be reduced to this choice if $T = 1$ (i.e., DirBN-1, with one layer only). However, unlike the sampling/optimising approaches used in Wallach et al. (2009), DirBN-1 uses a negative binomial augmentation shown in Eq. (3), which leads to a simpler inference scheme. Recently, models like Zhao et al. (2017a,c, 2018b) construct informative and asymmetric Dirichlet priors by taking into account some external knowledge like word embeddings. Whereas DirBN learns the asymmetric priors purely based on the context of the target corpus.

Instead of Dirichlet, the Pitman-Yor process (PYP) has been used on WDs to model the power-law distribution of words, as in Sato and Nakagawa (2010); Buntine and Mishra (2014). Chen et al. (2015) used a transformed PYP prior on $\phi$ to model multiple document collections. Lindsey et al. (2012) imposed a hierarchical PYP prior on $\phi$ to discover word phrases. Besides PYP, the Indian Buffet Process (IBP) has been used as a prior on $\phi$ to introduce word focusing on topics, as in Archambeau et al. (2015). In general, existing models use different priors on $\phi$ for modelling various linguistic phenomena, which have different purposes to DirBN. The deep structures induced by DirBN on WDs have not yet been rigorously studied.

To our knowledge, most existing models explore the structure of topics by imposing a deep/hierarchical prior on $\theta$. For example, hierarchical PYPs were used for domain adaptation in language models (Wood and Teh, 2009) and topic models (Du et al., 2012). nCRP (Blei et al., 2010) models topic hierarchies by introducing a tree-structured prior. Paisley et al. (2015); Kim et al. (2012); Ahmed et al. (2013) extended nCRP by either softening its constraints or applying it to different problems. Li and McCallum (2006) proposed the Pachinko Allocation model (PAM), which

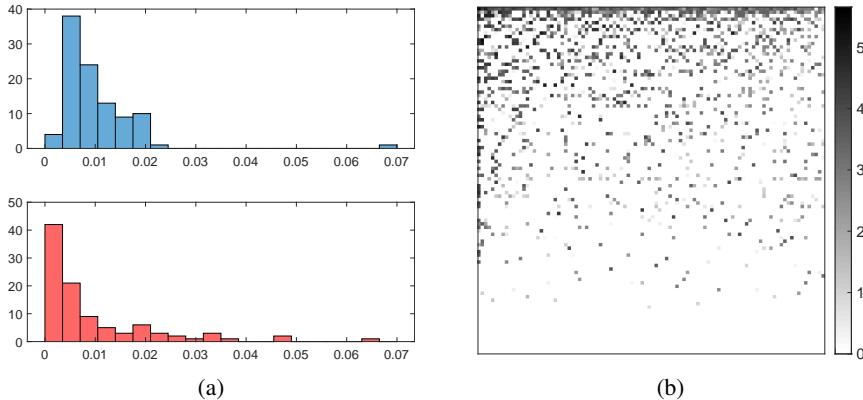

Figure 2: (a): Histograms of the normalised (latent) words counts. (b): $\mathbf{B}^{(1)}$.

captures the topic correlations with a directed acyclic graph. Recently, several deep extensions of PFA on $\theta$ have been proposed, including DPFA (Gan et al., 2015), DPFM (Henao et al., 2015), and GBN (Zhou et al., 2016). DPFM and GBN are the most related models to ours, which are also able to discover topic hierarchies. In DPFM and GBN, the higher-layer topics are not distributions over words but distributions over the topics in the layer below (they are called "meta-topics" in DPFM). To interpret those meta-topics, one needs to project them all the way down to the bottom-layer topics with matrix multiplication. Whereas in our model, the topics on all the layers are directly interpretable.

## 5  Experiments

The experiments were conducted on three real-world datasets, detailed as follows: **1)** Web Snippets (WS), containing 12,237 web search snippets labelled with 8 categories. The vocabulary contains 10,052 word types. **2)** Tag My News (TMN), consisting of 32,597 RSS news labelled with 7 categories. Each document contains a title and a description. There are 13,370 word types in the vocabulary. **3)** Twitter, extracted in 2011 and 2012 microblog tracks at Text REtrieval Conference (TREC)[5]. It has 11,109 tweets in total. The vocabulary size is 6,344.

With the framework of PFA, we compared three options of constructing $\phi$: (1) The default setting of PFA, where $\phi$ is drawn from a symmetric Dirichlet distribution with parameter 0.05, i.e., $\phi_k \sim \text{Dir}_V(0.05)$; (2) PFA+Mallet, where $\phi_k \sim \text{Dir}_V(\alpha_0)$ and $\alpha_0$ is sampled by Mallet [6]; (3) PFA+DirBN, the proposed model, where $\phi_k$ is drawn from an asymmetric Dirichlet distribution specific to $k$, the parameter of which is constructed with the higher-layer topics. Note that Wallach et al. (2009) tested the option using specific asymmetric Dirichlet parameter, i.e., $\phi_k \sim \text{Dir}_V([\alpha_1, \cdots, \alpha_V])$, but the performance is not as good as the symmetric parameter (the second one above). In addition, following a similar routine, we compared MetaLDA (Zhao et al., 2017a), and GBN (Zhou et al., 2016) with/without DirBN. Note that PFA is a widely used Bayesian topic model, MetaLDA is the state-of-the-art topic model capable of handling sparse texts, and GBN is reported Cong et al. (2017) to outperform many other deep models including DPFA (Gan et al., 2015), DPFM (Henao et al., 2015), nHDP (Paisley et al., 2015), and RSM (Hinton and Salakhutdinov, 2009).

For all the models, we ran 3,000 MCMC iterations with 1,500 burnin. For DirBN, we set $a_0 = b_0 = g_0 = h_0 = 1.0$ and $e_0 = f_0 = 0.01$. For PFA, MetaLDA, and GBN, we used their original implementations and settings, except that $\phi$ is drawn from DirBN in the combined models. For all the models, the number of topics in each layer of DirBN was set to 100, i.e., $K_T = \cdots = K_1 = 100$. For GBN and GBN+DirBN, we set the number of topics in each layer of GBN to 100 as well. Due to the shrinkage mechanisms of PFA, GBN, and DirBN, the number of active topics will be adjusted according to the data. In all the experiments, we varied the number of layers of DirBN $T$ from 1 to 3. For GBN+DirBN, the dual deep model, we fixed the number of layers of GBN as 3.

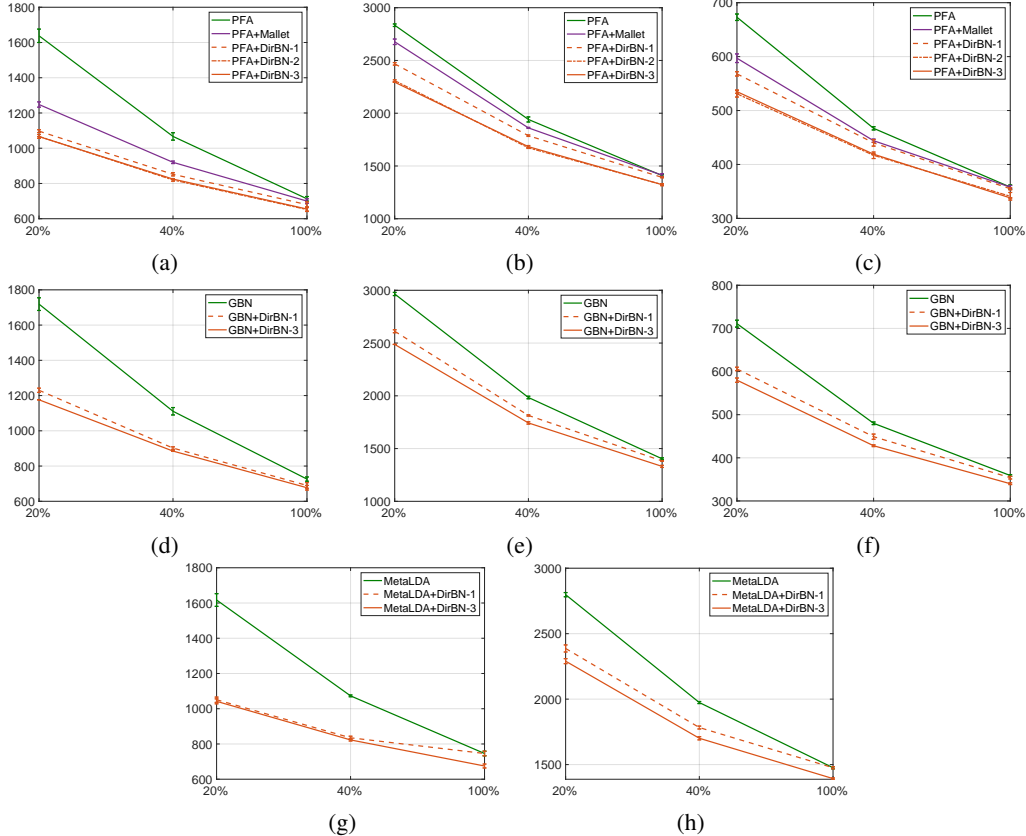

Figure 3: Perplexity (the vertical axis) with varied proportion (the horizontal axis) of the words for training in the training documents. (a-c): Results of the models based on PFA on WS, TMN, Twitter. (d-f): Results of the models based on GBN on WS, TMN, Twitter. (g,h): Results of the models based on MetaLDA on WS and TMN. The errorbars indicate the standard deviations of five runs. The number of a model indicates the number of layers used in DirBN. The results of MetaLDA and document classification on Twitter are not reported due to the unavailability of labels.

**Demonstration of DirBN's shrinkage effect**   As previously discussed, DirBN has an intrinsic shrinkage mechanism that is able to automatically learn the number of active topics in each layer (i.e., the network width). We empirically demonstrate the shrinkage effect in Figure 2, with the results of PFA+DirBN-3 on the TMN dataset. Figure 2a plots the histograms of the normalised (latent) words counts $\sum_v x_{vk_t}^{(t)} / \sum_{vk_t} x_{vk_t}^{(t)}$ for all $k_t$ where $x_{vk_t}^{(t)}$ is the word count for topic $k_t$. The blue and red bars are for the first- ($t = 1$) and the second-layer ($t = 2$) topics, respectively. The histogram indicates the number of topics ( the vertical axis) that are with a specific word count (the horizontal axis). A topic with a larger word count is more important. The shrinkage effect is that large proportion of the topics are with very small word counts, indicating that the number of effective topics is less than the truncation (i.e., $K_t = 100$). This is more obvious, in the second layer. Moreover, we display $\log \mathbf{B}^{(1)}$ as an image in Figure 2b. The vertical and horizontal axes are for the second- and first-layer topics, respectively. We ranked the first- and second-layer topics by their word counts. The sparsity of $\mathbf{B}^{(1)}$ indicates that the first- and second-layer topics are sparsely connected. This also demonstrates the shrinkage effect of the model.

**Quantitative results**   We report the per-heldout-word perplexity and topic coherence results. To compute perplexity, we randomly selected 80% of the documents in each dataset to train the models and 20% for testing. For each testing document, we randomly used one half of its words to infer its TP, and the other half to calculate perplexity. Topic coherence measures the semantic coherence in the most significant words (top words) of a topic. Here we used the Normalized Pointwise Mutual Information (NPMI) (Aletras and Stevenson, 2013; Lau et al., 2014) to calculate topic coherence

Table 1: Topic coherence with varied proportion of the words for training in the training documents. ± indicates the standard deviation of five runs. The best result in each column is in boldface.

| Training words | WS | | | TMN | | | Twitter | | |
|---|---|---|---|---|---|---|---|---|---|
| | 20% | 40% | 100% | 20% | 40% | 100% | 20% | 40% | 100% |
| PFA | -0.070±0.010 | 0.008±0.002 | 0.062±0.011 | -0.059±0.008 | 0.064±0.009 | 0.103±0.006 | -0.003±0.003 | 0.031±0.003 | 0.046±0.002 |
| PFA+Mallet | 0.008±0.004 | 0.049±0.005 | 0.063±0.003 | 0.035±0.006 | 0.083±0.005 | 0.108±0.005 | 0.022±0.003 | 0.037±0.002 | 0.045±0.003 |
| PFA+DirBN-1 | 0.013±0.003 | 0.052±0.004 | 0.060±0.006 | 0.031±0.003 | 0.080±0.001 | 0.108±0.008 | 0.019±0.004 | 0.037±0.004 | 0.049±0.007 |
| PFA+DirBN-3 | **0.021**±0.005 | 0.059±0.002 | 0.068±0.004 | 0.046±0.003 | 0.090±0.003 | 0.111±0.004 | 0.024±0.001 | 0.038±0.002 | 0.049±0.002 |
| GBN | -0.072±0.013 | 0.007±0.005 | 0.069±0.009 | -0.065±0.008 | 0.063±0.006 | 0.106±0.004 | -0.005±0.003 | 0.032±0.002 | 0.047±0.00 |
| GBN+DirBN-1 | 0.015±0.005 | 0.057±0.002 | 0.069±0.005 | 0.032±0.002 | 0.086±0.002 | 0.112±0.007 | 0.021±0.004 | 0.040±0.005 | 0.050±0.005 |
| GBN+DirBN-3 | 0.018±0.006 | **0.061**±0.004 | **0.075**±0.002 | 0.048±0.003 | **0.094**±0.004 | **0.113**±0.004 | **0.025**±0.003 | **0.040**±0.002 | **0.051**±0.003 |

Table 2: Topic hierarchy comparison in GBN+DirBN. Each row in boldface is the top 10 words in a first-layer topic. Each of these topics is associated with three most correlated topics in the second layer of DirBN (left) and GBN (right), respectively. The number associated with a second-layer topic is its (normalised) link weight to the first-layer topic.

| | DirBN | | GBN |
|---|---|---|---|
| **police arrested man charged woman authorities death year found accused** | | | |
| 0.13 | case charges accused trial courtattorney investigation judge allegations criminal | 0.38 | police arrested man charged year accused found charges woman death |
| 0.13 | police official killing attack deaddeath army security man family | 0.19 | police prison man china years arrested charges charged year chinese |
| 0.11 | woman men drug suicide girl sexual death found human york | 0.15 | china police chinese bomb fire people blast city artist officials |
| **heat miami james lebron game nba finals celtics bulls wade** | | | |
| 0.43 | season team game play run night star series fans career | 0.97 | heat miami james game nba finals lebron bulls mavericks dallas |
| 0.15 | nba playoffs court brink seeds defeated berth seed opponent semifinals | 0.00 | trial rajaratnam insider trading fund hedge raj anthony galleon case |
| 0.10 | win victory beat lead winning top fourth loss straight beating | 0.00 | music album lady gaga justin star pop band rock tour |
| **facebook google internet social twitter online web media site search** | | | |
| 0.18 | phone plan video technology mobile devices computer tech ceo content | 0.22 | facebook social internet google online twitter chief executive media web |
| 0.14 | company million buy billion corp industry sales companies consumers products | 0.19 | court lawsuit case facebook judge social federal internet google online |
| 0.12 | government report country nation pressure official state move released public | 0.18 | facebook social internet google online twitter world web media site |
| **study cancer drug risk heart patients women researchers disease people** | | | |
| 0.12 | rising percent high higher economic increase low growth strong recovery | 0.91 | study cancer drug risk researchers heart people patients health women |
| 0.12 | reactions periods technique method declared important realized treatment peril scores | 0.04 | world war years family oil dies year energy women american |
| 0.10 | study experience finding recent security kids challenges millions report special | 0.18 | facebook social internet google online twitter world web media site |
| **nuclear japan plant power radiation crisis japanese fukushima crippled tokyo** | | | |
| 0.17 | government united states officials state report country group official agency | 0.53 | nuclear japan plant power radiation crisis japanese fukushima earthquake tokyo |
| 0.14 | safety water nearby land found caused sea believed center parts | 0.44 | nuclear japan plant power radiation crisis japanese fukushima water tokyo |
| 0.13 | work plans part future system rules program bring offers decision | 0.01 | theater review broadway play york musical stage life time love |

score from the top 10 words of each topic and reported scores averaged over top 50 topics with highest NPMI, where "rubbish" topics are eliminated, following Yang et al. (2015)[7]. In the training documents, we further varied the proportion of the words used in training to mimic the case of sparse texts. All the models ran five times with different random seeds and we reported the averaged value with standard deviations.

The results of perplexity and topic coherence are shown in Figure 3 and Table 1, respectively. We have the following remarks on the results: **(1)** In general, for the models with DirBN, the performance is significantly improved compared with the counterparts without DirBN, especially in terms of perplexity and topic coherence and with low proportion of the training words. **(2)** In terms of all the measures, DirBN-2/3 always has better results than DirBN-1. Whereas if we compare GBN with PFA, its perplexity is worse than PFA's, especially for sparse texts. This demonstrates that hierarchical structures on $\theta$ (i.e., GBN) may not perform as well as hierarchical structures on $\phi$ (i.e., DirBN) on sparse texts. **(3)** Although PFA+DirBN-1 and PFA+Mallet both impose a symmetric Dirichlet on $\phi$, the former usually has better perplexity. **(4)** The dual deep model (GBN+DirBN-3) usually performs the best on topic coherence, which demonstrates the benefits of the deep structures.

**Qualitative analysis on topic hierarchies** [8] GBN+DirBN is a dual deep model that discovers two sets of hierarchies, one induced by GBN on $\theta$ and the other induced by DirBN on $\phi$. The topics in the

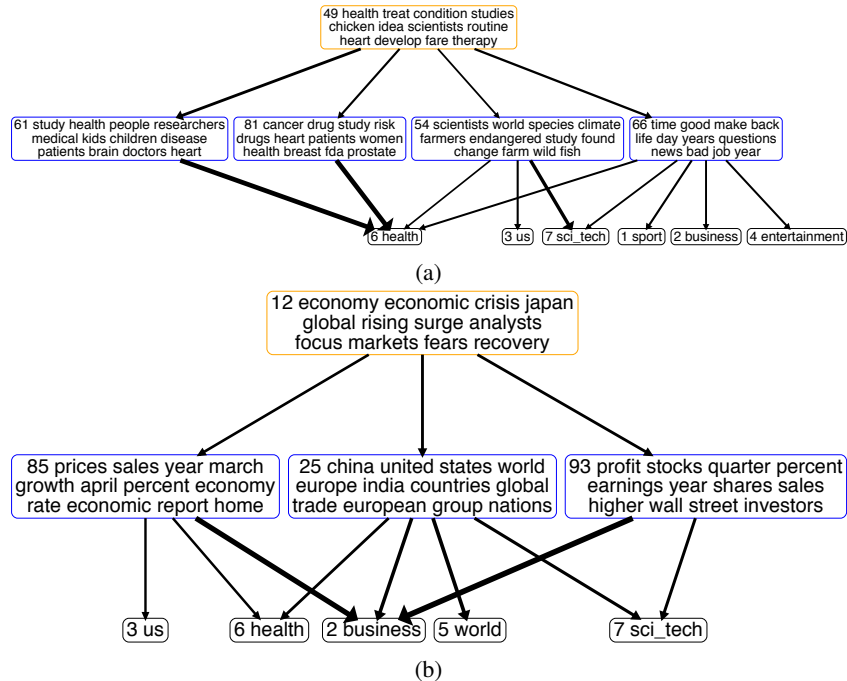

Figure 4: Topic hierarchies discovered by MetaLDA+DirBN. The topics in the yellow and blue rectangles are the second and first layer topics in DirBN and the correlated labels to the first-layer topics are shown at the bottom of each figure. Thicker arrows indicate stronger correlations.

first layer of DirBN connect the two sets of hierarchies. In Table 2, we show the first-layer topics and the correlated second-layer topics in the two hierarchies. It is interesting to see that the second-layer topics of DirBN are more abstract. For example, the second topic is about teams and player in NBA, while its correlated second-layer topics are more general words for sports. Moreover, DirBN is able to discover layer-wise semantically meaningful topic correlations with fewer overlapping top words. This is because GBN combines the words in the first-layer topics to form the second-layer topics, whereas DirBN decomposes the first-layer topics into the second-layer ones.

In MetaLDA+DirBN, the MetaLDA part is able to use document labels to construct TPs (Zhao et al., 2017a), by learning a correlation matrix between the labels and topics, while the DirBN part learns the topic hierarchy. The first-layer topics of DirBN link the correlation matrix and the topic hierarchy together. Figure 4 shows the sample linkages between topic hierarchies and labels on TMN, where the documents are labelled with 7 categories: 1 sport, 2 business, 3 us, 4 entertainment, 5 world, 6 health, 7 sci-tech. One can observe that there is a well correspondence between the topic hierarchies and the labels.

## 6    Conclusions

We have presented DirBN, a multi-layer process generating word distributions of topics. With real topics in each layer, DirBN is able to discover interpretable topic hierarchies. As a flexible module, DirBN can be adapted to other advanced topic models and improve the performance and interpretability, especially on sparse texts. We have demonstrated DirBN's advantages by equipping PFA, MetaLDA, and GBN, with DirBN. With the help of data augmentation, the inference of DirBN can be done by a layer-wise Gibbs sampling, as a full conjugate model.

Future directions include deriving alternative inference algorithms, such as variational inference (Hoffman et al., 2013), conditional density filtering (Guhaniyogi et al., 2018), and stochastic gradient-based approaches (Chen et al., 2014; Ding et al., 2014; Welling and Teh, 2011).

**Acknowledgments**

M. Zhou acknowledges the support of Award IIS-1812699 from the U.S. National Science Foundation.

## Footnotes

[2]Code available at `https://github.com/ethanhezhao/DirBN`

[3] $-$ can be a vector as a set of asymmetric parameters or a scalar as a symmetric parameter of Dirichlet

[4] Omitted details of inference as well as the overall algorithm are given in the supplementary materials.

[5]http://trec.nist.gov/data/microblog.html

[6]http://mallet.cs.umass.edu

[7]We used the Palmetto package (`http://palmetto.aksw.org`) with a large Wikipedia dump.

[8]More visualisations of topic hierarchies are shown in the supplementary material.

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
