[Supplementary Material]

# Supplementary materials for "Dirichlet belief networks for topic structure learning"

**He Zhao[1], Lan Du[1], Wray Buntine[1], and Mingyuan Zhou[2]**
[1]Faculty of Information Technology, Monash University, Australia
[2]McCombs School of Business, The University of Texas at Austin, USA

## 1 Details of the inference

Given the latent counts $\boldsymbol{x}_{k_t}^{(t)}$, the details of inference of the the $t$-th ($t < T$) layer of DirBN are as follows:

$$\boldsymbol{\phi}_{k_t}^{(t)} \sim \text{Dir}\left(\boldsymbol{\psi}_{k_t}^{(t)} + \boldsymbol{x}_{k_t}^{(t)}\right), \tag{1}$$

$$q_{k_1}^{(1)} \sim \text{Beta}\left(\psi_{\cdot k_1}^{(1)}, x_{\cdot k_1}^{(1)}\right), \tag{2}$$

$$y_{vk_t}^{(t)} \sim \text{CRT}\left(x_{vk_t}^{(t)}, \psi_{vk_t}^{(t)}\right), \tag{3}$$

$$\left(z_{v1k_t}^{(t)}, \cdots, z_{vK_{t+1}k_t}^{(t)}\right) \sim \text{Mult}\left(y_{vk_t}^{(t)}, \frac{\phi_{v1}^{(t+1)}\beta_{1k_t}^{(t)}}{\psi_{vk_t}^{(t)}}, \cdots, \frac{\phi_{vK_{t+1}}^{(t+1)}\beta_{K_{t+1}k_t}^{(t)}}{\psi_{vk_t}^{(t)}}\right), \tag{4}$$

$$\beta_{k_{t+1}k_t}^{(t)} \sim \text{Ga}\left(\gamma_{k_{t+1}}^{(t)} + z_{\cdot k_{t+1}k_t}^{(t)}, 1.0\right) / \left(c^{(t)} - \log q_{k_t}^{(t)}\right), \tag{5}$$

$$m_{k_{t+1}k_t}^{(t)} \sim \text{CRT}\left(z_{\cdot k_{t+1}k_t}^{(t)}, \gamma_{k_{t+1}}^{(t)}\right), \tag{6}$$

$$\gamma_{k_{t+1}}^{(t)} \sim \text{Ga}\left(\gamma_0^{(t)}/K_{t+1} + \sum_{k_t}^{K_t} m_{k_{t+1}k_t}^{(t)}, 1.0\right) / \left(c_0^{(t)} + n^{(t)}\right), \tag{7}$$

$$c^{(t)} \sim \text{Ga}\left(g_0 + K_t \sum_{k_{t+1}}^{K_{t+1}} \gamma_{k_{t+1}}^{(t)}, 1.0\right) / \left(h_0 + \sum_{k_{t+1},k_t}^{K_{t+1},K_t} \beta_{k_{t+1}k_t}^{(t)}\right), \tag{8}$$

$$p_{k_{t+1}}^{(t)} \sim \text{CRT}\left(\sum_{k_t}^{K_t} m_{k_{t+1}k_t}^{(t)}, \gamma_0^{(t)}/K_{t+1}\right), \tag{9}$$

$$\gamma_0^{(t)} \sim \text{Ga}\left(e_0 + \sum_{k_{t+1}}^{K_{t+1}} p_{k_{t+1}}^{(t)}, 1.0\right) / \left(f_0 + \log\frac{n^{(t)} + c_0^{(t)}}{c_0^{(t)}}\right), \tag{10}$$

$$c_0^{(t)} \sim \text{Ga}\left(g_0 + \gamma_0^{(t)}, 1.0\right) / \left(h_0 + \sum_{k_{t+1}} \gamma_{k_{t+1}}^{(t)}\right), \tag{11}$$

where $n^{(t)} = \sum_{k_t}^{K_t} \log\frac{c^{(t)} - \log q_{k_t}^{(t)}}{c^{(t)}}$.

In the top layer $t = T$, we have:

$$\phi_{k_T}^{(T)} \sim \text{Dir}\left(\eta + \boldsymbol{x}_{k_T}^{(T)}\right), \tag{12}$$

$$s_{vk_T} \sim \text{CRT}\left(x_{vk_T}^{(T)}, \eta\right), \tag{13}$$

$$\eta \sim \text{Ga}\left(a_0 + \sum_{v,k_T}^{V,K_T} s_{vK_T}, 1.0\right) / \left(b_0 - \sum_{k_T}^{K_T} \log q_{k_T}^{(T)}\right). \tag{14}$$

The inference process of DirBN is in Algorithm 1. Note that in different models, after the topic assignments of words are obtained, the inference of DirBN is the same.

## 2 Details of the combined models

**PFA+DirBN**    The generative process of PFA+DirBN is shown as follows:

$$p_k \sim \text{Beta}\left(c\epsilon, c(1-\epsilon)\right), r_k \sim \text{Ga}(c_0 r_0, 1/c_0), \theta_{kd} \sim \text{Ga}\left(r_k, \frac{p_k}{1-p_k}\right),$$

$$\phi_k \sim \text{DirBN}(T), x_{vd} = \sum_k^K x_{vdk}, x_{vdk} \sim \text{Pois}(\phi_{vk}\theta_{kd}), \tag{15}$$

where $\text{DirBN}(T)$ stands for the generative process of DirBN with $T$ layers.

**MetaLDA+DirBN**    The generative process of MetaLDA+DirBN is as follows:

$$\lambda_{lk} \sim \text{Ga}(a_0, 1/b_0), \alpha_{kd} = \prod_l^L (\lambda_{lk})^{f_{ld}}, \boldsymbol{\theta}_d \sim \text{Dir}(\boldsymbol{\alpha}_d), \phi_k \sim \text{DirBN}(T),$$

$$z_{id} \sim \text{Categorical}(\boldsymbol{\theta}_d), w_{id} \sim \text{Categorical}(\phi_{z_{id}}), \tag{16}$$

where $L$ is the number of unique document labels, $l \in \{1, \cdots, L\}$, $f_{ld} \in \{0, 1\}$ indicates whether document $d$ has label $l$, $w_{id} = v$ is the $i$-th word in document $d$, and $z_{id} = k$ is the topic assignment of $w_{id}$.

**GBN+DirBN**    The generative process of GBN+DirBN is as follows:

$$\boldsymbol{\theta}_d^{(S)} \sim \text{Ga}\left(\boldsymbol{r}, 1/c_j^{(T+1)}\right), \cdots, \boldsymbol{\theta}_d^{(s)} \sim \text{Ga}\left(\widetilde{\boldsymbol{\Phi}}^{(s+1)}\boldsymbol{\theta}_d^{(s+1)}, 1/c_d^{(s+1)}\right),$$

$$\cdots$$

$$\boldsymbol{\theta}_d^{(1)} \sim \text{Ga}\left(\widetilde{\boldsymbol{\Phi}}^{(2)}\boldsymbol{\theta}_d^{(2)}, p_d^{(2)}/(1-p_d^{(2)})\right), \phi_k \sim \text{DirBN}(T),$$

$$x_{vd} = \sum_k^K x_{vdk}, x_{vdk} \sim \text{Pois}(\phi_{vk}\theta_{kd}^{(1)}), \tag{17}$$

where $s \in \{1, \cdots, S\}$ is the index of the $s$ layer in GBN.

## 3 More results

For document classification, the TPs were used as input features for a $L_2$ regularized logistic regression using the LIBLINEAR package to predict the document labels. We used the same train/test splits as in perplexity evaluation, except that all the words in a test document were used to infer its TP. The results on WS and TMN are shown in Table 1.

## 4 Visualisation of topic hierarchies

Shown in Figure 2 to 5.

Table 1: Document classification

| Training words | WS | | | TMN | | |
|---|---|---|---|---|---|---|
| | 20% | 40% | 100% | 20% | 40% | 100% |
| PFA | 67.58±5.73 | 81.08±0.83 | 82.29±0.73 | 73.02±1.43 | 78.68±0.28 | 80.00±0.51 |
| PFA+mallet | 73.97±1.12 | 79.64±0.89 | 82.75±0.89 | 72.84±0.40 | 78.02±0.89 | 79.92±0.66 |
| PFA+DirBN-1 | 77.11±0.55 | 81.69±0.53 | 82.26±0.48 | 73.08±0.33 | 78.40±0.31 | 79.77±0.56 |
| PFA+DirBN-3 | 76.74±0.57 | 82.04±0.28 | 83.68±1.04 | 74.41±0.60 | 78.99±0.46 | 79.91±0.56 |
| MetaLDA | 67.94±3.00 | **83.26**±1.21 | 84.18±1.10 | 74.02±0.62 | 78.88±0.27 | 80.04±0.49 |
| MetaLDA + DirBN-1 | 76.67±0.88 | 81.38±1.02 | 83.07±0.70 | 74.10±0.22 | 79.67±0.67 | 80.63±0.10 |
| MetaLDA + DirBN-3 | 77.84±1.06 | 82.53±0.46 | 83.97±1.09 | 75.03±0.26 | 79.37±0.63 | 80.99±0.22 |
| GBN | 68.87±4.67 | 82.97±0.49 | 84.35±0.91 | 72.88±1.08 | 79.28±0.41 | **81.44**±0.21 |
| GBN+DirBN-1 | 76.73±0.70 | 82.54±0.81 | 83.18±0.40 | 74.42±0.32 | 79.59±0.30 | 80.87±0.68 |
| GBN+DirBN-3 | **78.17**±1.88 | 82.82±1.08 | **84.28**±1.12 | **75.36**±0.60 | **79.79**±0.48 | 81.10±0.34 |

**Require:** $\boldsymbol{x}_{k_1}^{(1)}$ for all $k_1, T(T>1), a_0, b_0, e_0, f_0, g_0, h_0 \ MaxIteration$
**Ensure:** $\boldsymbol{\beta}_{k_t}^{(t)}, \boldsymbol{\phi}_{k_t}^{(t)}$ for all $k_t$

1: Randomly initialise all the latent variables according to the generative process
2: **for** $iter \leftarrow 1$ **to** $MaxIteration$ **do**
3:     $/*$ Propagating the latent counts from the bottom up $*/$
4:     **for** $t \leftarrow 1$ **to** $T$ **do**
5:       **for all** $k_t$ and $v$ **do**
6:         Sample $y_{vk_t}^{(t)}$ by Eq. (3)
7:         **for all** $k_{t+1}$ **do**
8:           Sample $z_{vk_{t+1}k_t}^{(t)}$ by Eq. (4)
9:         **end for**
10:       **end for**
11:     **end for**
12:     $/*$ Updating the latent variables from the top down $*/$
13:     **for** $t \leftarrow T$ **to** $1$ **do**
14:       **if** $t = T$ **then**
15:         **for all** $k_T$ and $v$ **do**
16:           Sample $s_{vk_T}$ by Eq. (13)
17:         **end for**
18:         Sample $\eta$ by Eq. (14)
19:         **for all** $k_T$ **do**
20:           Sample $\boldsymbol{\phi}_{k_T}^{(T)}$ by Eq. (12)
21:         **end for**
22:       **else**
23:         **for all** $k_t$ **do**
24:           Compute $\boldsymbol{\psi}_{k_t}^{(t)}$ by $\boldsymbol{\psi}_{k_t}^{(t)} = \sum_{k_{t+1}}^{K_{t+1}} \boldsymbol{\phi}_{k_{t+1}}^{(t+1)} \beta_{k_{t+1}k_t}^{(t)}$
25:         **end for**
26:         **for all** $k_t$ **do**
27:           Sample $q_{k_t}^{(t)}$ by Eq. (2)
28:         **end for**
29:         **for all** $k_t$ and $k_{t+1}$ **do**
30:           Sample $m_{k_{t+1}k_t}^{(t)}$ by Eq. (6)
31:         **end for**
32:         **for all** $k_{t+1}$ **do**
33:           Sample $\gamma_{k_{t+1}}^{(t)}, p_{k_{t+1}}^{(t)}$ by Eq. (7,9)
34:         **end for**
35:         Sample $c^{(t)}, \gamma_0^{(t)}, c_0^{(t)}$ by Eq. (8,10,11)
36:         **for all** $k_t$ and $k_{t+1}$ **do**
37:           Sample $\beta_{k_{t+1}k_t}^{(t)}$ by Eq. (5)
38:         **end for**
39:         **for all** $k_t$ **do**
40:           Sample $\boldsymbol{\phi}_{k_t}^{(t)}$ by Eq. (1)
41:         **end for**
42:       **end if**
43:     **end for**
44: **end for**

Figure 1: Infernece algorithm for DirBN

Figure 2: Topic hierarchies discovered by MetaLDA+DirBN on TMN. The topics in the green, yellow, and blue rectangles are the third, second, and first layer topics in DirBN and the correlated document labels are shown on the bottom of each figure. Thicker arrows indicate stronger correlations.

Figure 3: Topic hierarchies discovered by MetaLDA+DirBN on WS. The topics in the green, yellow, and blue rectangles are the third, second, and first layer topics in DirBN and the correlated document labels are shown on the bottom of each figure. Thicker arrows indicate stronger correlations.

Figure 4: Topic hierarchies discovered by GBN+DirBN on TMN. The topics in the red and green rectangles are the third and second-layer topics discovered by GBN on TPs. The topics in the blue rectangles are the second-layer topics discovered by DirBN on WDs. The topics in the yellow rectangles are the first-layer topics connecting the higher-layer topics of GBN and DirBN.

Figure 5: Topic hierarchies discovered by GBN+DirBN on WS. The topics in the red and green rectangles are the third and second-layer topics discovered by GBN on TPs. The topics in the blue rectangles are the second-layer topics discovered by DirBN on WDs. The topics in the yellow rectangles are the first-layer topics connecting the higher-layer topics of GBN and DirBN.