[Reviews · NeurIPS 2018]

Reviewer 1



This submission proposes a new prior on the topic-word distribution in latent topic models. This model defines a multi-layer feedforward graph, where each layer contains a set of valid multinomial distributions over the vocabulary, and weighted combinations of each layer's "topics" are used as the Dirichlet prior for the "topics" of the next layer. The key purported benefits are sharing of statistical strengh, inference of a hierarchy of interpretable "abstract" topics, and modularity that allows composition with other topic model variants that modify the document-topic distributions. The authors present an efficient fully collapsed Gibbs sampler inference scheme - I did not thoroughly check the derivation but it seems plausible. Although: what is the computational complexity (and relative "wall clock" cost) of the given inference scheme? The experimental datasets used seem relatively small. The quantitative evaluation shows improved results on typical topic modeling metrics: perplexity, coherence, and auxiliary classification task accuracy. A handful of chosen topic hiearchies suggest nice "abstraction" in the parent topics, although I would have liked to see some quantitative analysis of this aspect as well. Exploiting the compositionality with document-topic variants for the experiments was a nice touch that I liked. I was curious how sensitive the results were to variation in the hierarchy depthy or width. Some data about the inferred connectivity weights between layers would have been interesting as well (relative sparsity, concentation of weight, etc). Overall I found this paper to be an original, high-quality contribution that should admit further extensions and combinations. The paper is clearly written thoughout, well-contexualized with respect to the relevant literature, and the core idea is simple (in a good way). I found the topic hierarchy diagrams in Figure 2 (and the supplement) to be too small to easily scan and read. L217: can the correlation between the the topic and labels be quantified somehow? UPDATE: I have read author feedback and found the answers to my questions useful. Space permitting, it would be nice for some versions of those answers to be incorporated into the final submission.

Reviewer 2



The paper introduces the Dirichlet Belief Net as a general and deeper drop-in replacement for Dirichlet prior distributions for multinomial data. The author(s) derive an elegant data augmented Gibbs sampler to sample the parameters in what they call the Dirichlet Belief Net. They also give three application examples where this prior can replace standard Dirichlet distribution models, such as in Poisson Factor Analysis, MetaLDA, and the Gamma Belief Networks. The work is motivated by the fact that in the three mentioned models, previous work in deep topic modeling has not been focused on deep structures in the word-topic distribution. I’m no expert in the field of Deep Belief Net, so I’m not sure that similar approaches have not been proposed previously. I’m also not sure exactly how big a contribution this is compared to other similar approaches. But I think the idea is both novel, general and the proposed data augmented Gibbs sampling algorithm make the paper an attractive contribution that could be studied and developed further in other model settings. The proposed approach also seems to produce a better perplexity as well as better topic coherence. Quality: The technical content of the paper appears to be correct and the experimental result is convincing that this is a promising approach. There may be a couple of minor typos in the Inference section in the paper:: i) At line 115, I suspect that $q^\beta_{\cdot k}$ should be $q^\psi_{\cdot k_1}}$ ii) At line 116, can it be so that $z_{vk_{2}k_{1}}^{(1)}$ should be $\sum_{k_{2}}^{K_{2}}z_{vk_{2}k_{1}}^{(1)}=y_{vk_{1}}^{(1)}$? That would make more sense for me, but I may be wrong. One issue is the question of using documentation classification as a method for evaluating topic models. The documentation classification does not show any significant results with the new DirBN. I wonder about the argument of including this as an evaluation metric in the paper. Why would we expect a better classification accuracy if we have a better WP part of the model? I think this should be explained and/or argued for in the paper. Clarity: The paper is, in general, is well written, I think there are some parts that can be improved to make it easier and more clear for the reader. As an example, I find the Related Work being partly overlapping with the introduction. As an example, see paragraph 2 in the introduction and the last paragraph of the related work section. Figure 1 was initially a little difficult to follow and understand, but after a while, I think it was quite informative. But maybe the author(s) could add an explanation in the caption, helping the reader to understand the Figure easier, maybe by explaining what the three slides are, since they are different things, even though they look similar. There are parts in Table 1 that are not explained anywhere in the text, such as the different percentage of the different corpora (I guess this is the percentage of the corpus used as training set?). This should be explained in the paper or in the title caption. Also, I guess that the +- indicate the standard deviation of the different estimates. This should also be stated together with how this was computed in the experiment section. This is not possible to follow or reproduce in the current version. Minor things: i) At line 80, define what k_{t+1} is in a similar way as k is defined previously. ii) There are also errors in the boldfacing of the results in Table 1 (a) I think MetaLDA DirBN3 has better performance than the boldfaced results for WS20% and TM20%. Originality: Since I’m not an expert in the field of deep probabilistic models, I am not sure if this work has been done before. But the idea of Dirichlet distributed hidden layers seem to be a good idea and the results are promising. Significance: The experimental results using the DirBN is promising. Including the DirBN in three different models all improve the model perplexity and topic coherence. The only question is how well the approach would work compared with other similar approaches that has been used to do deep modeling of TP. The straight-forward MCMC sampling scheme using local conjugacy makes the approach promising and straightforward to use in other models, this in turn make the approach an attractive contribution. The approach can easily be incorporated in other models, studied and developed further. UPDATE AFTER REBUTTAL: I think most of my issues were explained. But it did not clarify \emph{why} DPN for WP should increase classification accuracy. Why would we expect this to happen? But this is a minor issue.

Reviewer 3



The paper introduces a deep belief net architecture for re-expressing the probability distribution of words in a topic hierarchically for deep topic model architectures. For each belief layer, the authors use the product of a sample from a gamma distribution to reweight the topic probabilities from the previous layer as a Dirichlet prior for the new layer's topic probabilities - hence the name, Dirichlet Belief Net (DirBN). The authors incorporate this system with Gibbs sampling for inference into a variety of existing models to demonstrate its flexibility and the improvement in both perplexity and coherence it provides to topic models. The authors do a good job of motivating this model, grounding it in existing work, providing implementation information and showing DirBN improves on these models. I think the perplexity, coherence, and document classification evaluations are appropriate, as are the choice of models to evaluate. I have a bunch of smaller points to discuss for the paper, but most of them have to do with clarity. One question lingering after reading this paper, though, relates to the "hierarchy" of topics being learned. The model consistently uses the same number of topics per layer, so it is unclear to me that each layer is offering higher-level intuitions so much as just offering alternative ways of organizing the information. The qualitative analysis provided by the authors did not do much to convince me otherwise; I am not sure that preventing overlap in top topic terms in some examples maps to a meaningful improvement in final topic quality. I also was less interested in which model was best across all options than whether DirBN improved over the baseline model in a statistically significant way for each of PFA, MetaLDA, and GBN, which seems to be true in several cases but is hard to read off of Tables 1(a-c) right now. I was also a little curious about whether the authors would expect this work to scale well to larger vocabularies, as most of the corpora tested are modest compared to some topic models of e.g. Wikipedia. My last technical concern relates to the perplexity measurement, as line 182 suggests it is deviating a bit from what I would expect for left-to-right estimation for topic model perplexity via "Evaluation Methods for Topic Models" (Wallach et al. 2009). Is there a particular reason for the deviation? Moving towards my clarity concerns: some of the structure of the paper gets confusing. My suggestion would be to better streamline the discussion in the paper by placing the discussion of inference for DirBN (section 3) between the original description of the model and the descriptions of inference for models incorporating DirBN (both section 2). It might also be good to explicitly separate out the final inference steps people would use (the "To summarize" section) into an algorithm or an enumerated list of sampling steps for different variables; the quantity of inline math in this explanation makes it hard to navigate. More of the math could probably also be moved to the appendix, aside from mentioning the auxiliary variables used and the sampling distributions for x, y, and z. I have a bunch of more minor corrections as well: - line 46: drawn from Dirichlet distributions - line 55: to a document, whereas our model (the sentence beginning with Whereas isn't a complete sentence right now) - line 64: applying DirBN to several well-developed models, including Poisson Factor Analysis - line #? (just above equation 1) - probably "vanilla LDA", not "a vanilla LDA" - rather than providing the ellipsis version of the initializations in equation 1, just putting the first two lines without ellipses would probably suffice) - line 159: you only list three datasets - line 162: tokens are instances of words, probably better to say "There are 13,370 types in the vocabulary" I think this paper is meaningful, well-balanced work that contributes to the deep learning for topic modeling literature. I would highly recommend modifications to improve clarity and address some of my outstanding technical questions for a final version. AFTER REVIEW: The authors did a good job of addressing my outstanding questions, particularly about the topic hierarchy and the vocabulary scaling. I hope the authors will include some explanation of that and the perplexity evaluation in the paper.